# Differentially Private Learning of Structured Discrete Distributions

**Ilias Diakonikolas**[*]
University of Edinburgh

**Moritz Hardt**
Google Research

**Ludwig Schmidt**
MIT

## Abstract

We investigate the problem of learning an unknown probability distribution over a discrete population from random samples. Our goal is to design efficient algorithms that simultaneously achieve low error in total variation norm while guaranteeing Differential Privacy to the individuals of the population.

We describe a general approach that yields near sample-optimal and computationally efficient differentially private estimators for a wide range of well-studied and natural distribution families. Our theoretical results show that for a wide variety of structured distributions there exist private estimation algorithms that are nearly as efficient—both in terms of sample size and running time—as their non-private counterparts. We complement our theoretical guarantees with an experimental evaluation. Our experiments illustrate the speed and accuracy of our private estimators on both synthetic mixture models and a large public data set.

## 1 Introduction

The majority of available data in modern machine learning applications come in a raw and unlabeled form. An important class of unlabeled data is naturally modeled as samples from a probability distribution over a *very large* discrete domain. Such data occurs in almost every setting imaginable—financial transactions, seismic measurements, neurobiological data, sensor networks, and network traffic records, to name a few. A classical problem in this context is that of *density estimation* or *distribution learning*: Given a number of iid samples from an unknown target distribution, we want to compute an accurate approximation of the distribution. *Statistical* and *computational* efficiency are the primary performance criteria for a distribution learning algorithm. More specifically, we would like to design an algorithm whose sample size requirements are information-theoretically optimal, and whose running time is nearly linear in its sample size.

Beyond computational and statistical efficiency, however, data analysts typically have a variety of additional criteria they must balance. In particular, data providers often need to maintain the anonymity and privacy of those individuals whose information was collected. *How can we reveal useful statistics about a population, while still preserving the privacy of individuals?* In this paper, we study the problem of density estimation in the presence of *privacy constraints*, focusing on the notion of differential privacy [1].

**Our contributions.** Our main findings suggest that the marginal cost of ensuring differential privacy in the context of distribution learning is only moderate. In particular, for a broad class of shape-constrained density estimation problems, we give private estimation algorithms that are nearly as efficient—both in terms of sample size and running time—as a nearly optimal *non-private* baseline. As our learning algorithm approximates the underlying distribution up to small error in total variation norm, all crucial properties of the underlying distribution are preserved. In particular, the analyst is free to compose our learning algorithm with an *arbitrary* non-private analysis.

---

[*]The authors are listed in alphabetical order.

Our strong positive results apply to all distribution families that can be well-approximated by piecewise polynomial distributions, extending a recent line of work [2, 3, 4] to the differentially private setting. This is a rich class of distributions including several natural mixture models, log-concave distributions, and monotone distributions amongst many other examples. Our algorithm is *agnostic* so that even if the unknown distribution does not conform exactly to any of these distribution families, it continues to find a good approximation.

These surprising positive results stand in sharp contrast with a long line of worst-case hardness results and lower bounds in differential privacy, which show separations between private and non-private learning in various settings.

Complementing our theoretical guarantees, we present a novel heuristic method to achieve empirically strong performance. Our heuristic always guarantees privacy and typically converges rapidly. We show on various data sets that our method scales easily to input sizes that were previously prohibitive for any implemented differentially private algorithm. At the same time, the algorithm approaches the estimation error of the best known non-private method for a sufficiently large number of samples.

**Technical overview.** We briefly introduce a standard model of learning an unknown probability distribution from samples (namely, that of [5]), which is essentially equivalent to the minimax rate of convergence in $\ell_1$-distance [6]. A distribution learning problem is defined by a class $\mathcal{C}$ of distributions. The algorithm has access to independent samples from an unknown distribution $p$, and its goal is to output a hypothesis distribution $h$ that is "close" to $p$. We measure the closeness between distributions in *total variation distance*, which is equivalent to the $\ell_1$-distance and sometimes also called *statistical distance*. In the "noiseless" setting, we are promised that $p \in \mathcal{C}$, and the goal is to construct a hypothesis $h$ such that (with high probability) the total variation distance $d_{\mathrm{TV}}(h, p)$ between $h$ and $p$ is at most $\alpha$, where $\alpha > 0$ is the accuracy parameter.

The more challenging "noisy" or *agnostic* model captures the situation of having arbitrary (or even adversarial) noise in the data. In this setting, we do not make any assumptions about the target distribution $p$ and the goal is to find a hypothesis $h$ that is almost as accurate as the "best" approximation of $p$ by any distribution in $\mathcal{C}$. Formally, given sample access to a (potentially arbitrary) target distribution $p$ and $\alpha > 0$, the goal of an *agnostic learning algorithm for $\mathcal{C}$* is to compute a hypothesis distribution $h$ such that $d_{\mathrm{TV}}(h, p) \leq C \cdot \mathrm{opt}_{\mathcal{C}}(p) + \alpha$, where $\mathrm{opt}_{\mathcal{C}}(p)$ is the total variation distance between $p$ and the closest distribution to it in $\mathcal{C}$, and $C \geq 1$ is a universal constant.

It is a folklore fact that learning an arbitrary discrete distribution over a domain of size $N$ to constant accuracy requires $\Omega(N)$ samples and running time. The underlying algorithm is straightforward: output the empirical distribution. For distributions over very large domains, a linear dependence on $N$ is of course impractical, and one might hope that drastically better results can be obtained for most natural settings. Indeed, there are many natural and fundamental distribution estimation problems where significant improvements are possible. Consider for example the class of all *unimodal* distributions over $[N]$. In sharp contrast to the $\Omega(N)$ lower bound for the unrestricted case, an algorithm of Birgé [7] is known to learn any unimodal distribution over $[N]$ with running time and sample complexity of $O(\log(N))$.

The starting point of our work is a recent technique [3, 8, 4] for learning univariate distributions via piecewise polynomial approximation. *Our first main contribution is a generalization of this technique to the setting of approximate differential privacy.* To achieve this result, we exploit a connection between structured distribution learning and private "Kolmogorov approximations". More specifically, we show in Section 3 that, for the class of structured distributions we consider, a private algorithm that approximates an input histogram in the *Kolmogorov distance* combined with the algorithmic framework of [4] yields sample and computationally efficient private learners under the *total variation distance*.

Our approach crucially exploits the structure of the underlying distributions, as the Kolmogorov distance is a much weaker metric than the total variation distance. Combined with a recent private algorithm [9], we obtain differentially private learners for a wide range of structured distributions over $[N]$. The sample complexity of our algorithms matches their non-private analogues up to a standard dependence on the privacy parameters and a multiplicative factor of at most $O(2^{\log^* N})$,

where $\log^*$ denotes the iterated logarithm function. The running time of our algorithm is nearly-linear in the sample size and *logarithmic* in the domain size.

**Related Work.** There is a long history of research in statistics on estimating structured families of distributions going back to the 1950's [10, 11, 12, 13], and it is still a very active research area [14, 15, 16]. Theoretical computer scientists have also studied these problems with an explicit focus on the computational efficiency [5, 17, 18, 19, 3]. In statistics, the study of inference questions under privacy constraints goes back to the classical work of Warner [20]. Recently, Duchi *et al.* [21, 22] study the trade-off between statistical efficiency and privacy in a *local* model of privacy obtaining sample complexity bounds for basic inference problems. We work in the non-local model and our focus is on both statistical and computational efficiency.

There is a large literature on answering so-called "range queries" or "threshold queries" over an ordered domain subject to differential privacy. See, for example, [23] as well as the recent work [24] and many references therein. If the output of the algorithm represents a histogram over the domain that is accurate on all such queries, then this task is equivalent to approximating a sample in Kolmogorov distance, which is the task we consider. Apart from the work of Beimel et al. [25] and Bun et al. [9], to the best of our knowledge all algorithms in this literature (e.g., [23, 24]) have a running time that depends polynomially on the domain size $N$. Moreover, except for the aforementioned works, we know of no other algorithm that achieves a sub-logarithmic dependence on the domain size in its approximation guarantee. Of all algorithms in this area, we believe that ours is the first implemented algorithm that scales to very large domains with strong empirical performance as we demonstrate in Section 5.

# 2 Preliminaries

**Notation and basic definitions.** For $N \in \mathbb{Z}_+$, we write $[N]$ to denote the set $\{1, \ldots, N\}$. The $\ell_1$-norm of a vector $v \in \mathbb{R}^N$ (or equivalently, a function from $[N]$ to $\mathbb{R}$) is $\|v\|_1 = \sum_{i=1}^{N} |v_i|$. For a discrete probability distribution $p : [N] \to [0, 1]$, we write $p(i)$ to denote the probability of element $i \in [N]$ under $p$. For a subset of the domain $S \subseteq [N]$, we write $p(S)$ to denote $\sum_{i \in S} p(i)$. The *total variation distance* between two distributions $p$ and $q$ over $[N]$ is $d_{TV}(p, q) \stackrel{\text{def}}{=} \max_{S \subseteq [N]} |p(S) - q(S)| = (1/2) \cdot \|p - q\|_1$. The *Kolmogorov distance* between $p$ and $q$ is defined as $d_{\mathrm{K}}(p, q) \stackrel{\text{def}}{=} \max_{j \in [N]} |\sum_{i=1}^{j} p(i) - \sum_{i=1}^{j} q(i)|$. Note that $d_{\mathrm{K}}(p, q) \leq d_{TV}(p, q)$. Given a set $S$ of $n$ independent samples $s_1, \ldots, s_n$ drawn from a distribution $p : [N] \to [0, 1]$, the *empirical distribution* $\widehat{p}_n : [N] \to [0, 1]$ is defined as follows: for all $i \in [N]$, $\widehat{p}_n(i) = |\{j \in [n] \mid s_j = i\}| / n$.

**Definition 1** (Distribution Learning). *Let $\mathcal{C}$ be a family of distributions over a domain $\Omega$. Given sample access to an unknown distribution $p$ over $\Omega$ and $0 < \alpha, \beta < 1$, the goal of an $(\alpha, \beta)$-agnostic learning algorithm for $\mathcal{C}$ is to compute a hypothesis distribution $h$ such that with probability at least $1 - \beta$ it holds $d_{TV}(h, p) \leq C \cdot \mathrm{opt}_{\mathcal{C}}(p) + \alpha$ , where $\mathrm{opt}_{\mathcal{C}}(p) := \inf_{q \in \mathcal{C}} d_{TV}(q, p)$ and $C \geq 1$ is a universal constant.*

**Differential Privacy.** A *database* $D \in [N]^n$ is an $n$-tuple of items from $[N]$. Given a database $D = (d_1, \ldots, d_n)$, we let $\mathrm{hist}(D)$ denote the normalized histogram corresponding to $D$. That is, $\mathrm{hist}(D) = \frac{1}{n} \sum_{i=1}^{n} \mathbf{e}_{d_i}$, where $\mathbf{e}_j$ denotes the $j$-th standard basis vector in $\mathbb{R}^N$.

**Definition 2** (Differential Privacy). *A randomized algorithm $M : [N]^n \to \mathcal{R}$ (where $\mathcal{R}$ is some arbitrary range) is $(\epsilon, \delta)$-differentially private if for all pairs of inputs $D, D' \in [N]^n$ differing in only one entry, we have that for all subsets of the range $S \subseteq \mathcal{R}$, the algorithm satisfies:*

$$\Pr[M(D) \in S] \leq \exp(\epsilon) \Pr[M(D') \in S] + \delta.$$

In the context of private distribution learning, the database $D$ is the sample set $S$ from the unknown target distribution $p$. In this case, the normalized histogram corresponding to $D$ is the same as the empirical distribution corresponding to $S$, i.e., $\mathrm{hist}(S) = \widehat{p}_n(S)$.

**Basic tools from probability.** We recall some probabilistic inequalities that will be crucial for our analysis. Our first tool is the well-known *VC inequality*. Given a family of subsets $\mathcal{A}$ over $[N]$, define $\|p\|_{\mathcal{A}} = \sup_{A \in \mathcal{A}} |p(A)|$. The *VC–dimension* of $\mathcal{A}$ is the maximum size of a subset $X \subseteq [N]$ that is shattered by $\mathcal{A}$ (a set $X$ is shattered by $\mathcal{A}$ if for every $Y \subseteq X$ some $A \in \mathcal{A}$ satisfies $A \cap X = Y$).

**Theorem 1** (VC inequality, [6, p. 31]). *Let $\widehat{p}_n$ be an empirical distribution of $n$ samples from $p$. Let $\mathcal{A}$ be a family of subsets of VC–dimension $k$. Then $\mathbb{E}\left[\|p - \widehat{p}_n\|_{\mathcal{A}}\right] \leq O(\sqrt{k/n})$.*

We note that the RHS above is best possible (up to constant factors) and independent of the domain size $N$. The *Dvoretzky-Kiefer-Wolfowitz (DKW) inequality* [26] can be obtained as a consequence of the VC inequality by taking $\mathcal{A}$ to be the class of all intervals. The DKW inequality implies that for $n = \Omega(1/\epsilon^2)$, with probability at least $9/10$ (over the draw of $n$ samples from $p$) the empirical distribution $\widehat{p}_n$ will be $\epsilon$-close to $p$ in Kolmogorov distance.

We will also use the following uniform convergence bound:

**Theorem 2** ([6, p. 17]). *Let $\mathcal{A}$ be a family of subsets over $[N]$, and $\widehat{p}_n$ be an empirical distribution of $n$ samples from $p$. Let $X$ be the random variable $\|p - \hat{p}\|_{\mathcal{A}}$. Then we have $\Pr\left[X - \mathbb{E}[X] > \eta\right] \leq e^{-2n\eta^2}$.*

**Connection to Synthetic Data.** Distribution learning is closely related to the problem of generating synthetic data. Any dataset $D$ of size $n$ over a universe $X$ can be interpreted as a distribution over the domain $\{1, \ldots, |X|\}$. The weight of item $x \in X$ corresponds to the fraction of elements in $D$ that are equal to $x$. In fact, this histogram view is convenient in a number of algorithms in Differential Privacy. If we manage to learn this unknown distribution, then we can take $n$ samples from it obtain another *synthetic* dataset $D'$. Hence, the quality of the distribution learner dictates the statistical properties of the synthetic dataset. Learning in total variation distance is particularly appealing from this point of view. If two datasets represented as distributions $p, q$ satisfy $d_{\mathrm{TV}}(p, q) \leq \alpha$, then for *every* test function $f : X \rightarrow \{0, 1\}$ we must have that $|\mathbb{E}_{x \sim p} f(x) - \mathbb{E}_{x \sim q} f(x)| \leq \alpha$. Put in different terminology, this means that the answer to any statistical query[1] differs by at most $\alpha$ between the two distributions.

## 3 A Differentially Private Learning Framework

In this section, we describe our private distribution learning upper bounds. We start with the simple case of privately learning an arbitrary discrete distribution over $[N]$. We then extend this bound to the case of privately and agnostically learning a histogram distribution over an arbitrary but known partition of $[N]$. Finally, we generalize the recent framework of [4] to obtain private agnostic learners for histogram distributions over an arbitrary unknown partition, and more generally piecewise polynomial distributions.

Our first theorem gives a differentially private algorithm for arbitrary distributions over $[N]$ that essentially matches the best non-private algorithm. Let $\mathcal{C}_N$ be the family of all probability distributions over $[N]$. We have the following:

**Theorem 3.** *There is a computationally efficient $(\epsilon, 0)$-differentially private $(\alpha, \beta)$-learning algorithm for $\mathcal{C}_N$ that uses $n = O((N + \log(1/\beta))/\alpha^2 + N \log(1/\beta)/(\epsilon\alpha))$ samples.*

The algorithm proceeds as follows: Given a dataset $S$ of $n$ samples from the unknown target distribution $p$ over $[N]$, it outputs the hypothesis $h = \mathrm{hist}(S) + \eta = \widehat{p}_n(S) + \eta$, where $\eta \in \mathbb{R}^N$ is sampled from the $N$-dimensional Laplace distribution with standard deviation $1/(\epsilon n)$. The simple analysis is deferred to Appendix A.

A *$t$-histogram* over $[N]$ is a function $h : [N] \rightarrow \mathbb{R}$ that is piecewise constant with at most $t$ interval pieces, i.e., there is a partition $\mathcal{I}$ of $[N]$ into intervals $I_1, \ldots, I_t$ such that $h$ is constant on each $I_i$. Let $\mathcal{H}_t(\mathcal{I})$ be the family of all $t$-histogram distributions over $[N]$ with respect to partition $\mathcal{I} = \{I_1, \ldots, I_t\}$. Given sample access to a distribution $p$ over $[N]$, our goal is to output a hypothesis $h : [N] \rightarrow [0, 1]$ that satisfies $d_{\mathrm{TV}}(h, p) \leq C \cdot \mathrm{opt}_t(p) + \alpha$, where $\mathrm{opt}_t(p) = \inf_{g \in \mathcal{H}_t(\mathcal{I})} d_{\mathrm{TV}}(g, p)$. We show the following:

**Theorem 4.** *There is a computationally efficient $(\epsilon, 0)$-differentially private $(\alpha, \beta)$-agnostic learning algorithm for $\mathcal{H}_t(\mathcal{I})$ that uses $n = O((t + \log(1/\beta))/\alpha^2 + t \log(1/\beta)/(\epsilon\alpha))$ samples.*

The main idea of the proof is that the differentially private learning problem for $\mathcal{H}_t(\mathcal{I})$ can be reduced to the same problem over distributions of support $[t]$. The theorem then follows by an

application of Theorem 3. See Appendix A for further details. Theorem 4 gives differentially private learners for any family of distributions over $[N]$ that can be well-approximated by histograms over a fixed partition, including monotone distributions and distributions with a known mode.

In the remainder of this section, we focus on approximate privacy, i.e., $(\epsilon, \delta)$-differential privacy for $\delta > 0$, and show that for a wide range of natural and well-studied distribution families there exists a computationally efficient and differentially private algorithm whose sample size is at most a factor of $2^{O(\log^* N)}$ worse than its non-private counterpart. In particular, we give a differentially private version of the algorithm in [4]. For a wide range of distributions, our algorithm has near-optimal sample complexity and runs in time that is *nearly-linear* in the sample size and *logarithmic* in the domain size.

We can view our overall private learning algorithm as a *reduction*. For the sake of concreteness, we state our approach for the case of histograms, the generalization to piecewise polynomials being essentially identical. Let $\mathcal{H}_t$ be the family of all $t$-histogram distributions over $[N]$ (with unknown partition). In the non-private setting, a combination of Theorems 1 and 2 (see appendix) implies that $\mathcal{H}_t$ is $(\alpha, \beta)$-agnostically learnable using $n = \Theta((t + \log(1/\beta))/\alpha^2)$ samples. The algorithm of [4] starts with the empirical distribution $\widehat{p}_n$ and post-processes it to obtain an $(\alpha, \beta)$-accurate hypothesis $h$. Let $\mathcal{A}_k$ be the collection of subsets of $[N]$ that can be expressed as unions of at most $k$ disjoint intervals. The important property of the empirical distribution $\widehat{p}_n$ is that with high probability, $\widehat{p}_n$ is $\alpha$-close to the target distribution $p$ in $\mathcal{A}_k$-distance for any $k = O(t)$.

The crucial observation that enables our generalization is that *the algorithm of [4] achieves the same performance guarantees starting from any hypothesis $q$ such that $\|p - q\|_{\mathcal{A}_{O(t)}} \leq \alpha$.*[2] This observation motivates the following two-step differentially private algorithm: (1) Starting from the empirical distribution $\widehat{p}_n$, efficiently construct an $(\epsilon, \delta)$-differentially private hypothesis $q$ such that with probability at least $1 - \beta/2$ it holds $\|q - \widehat{p}_n\|_{\mathcal{A}_{O(t)}} \leq \alpha/2$. (2) Pass $q$ as input to the learning algorithm of [4] with parameters $(\alpha/2, \beta/2)$ and return its output hypothesis $h$.

We now proceed to sketch correctness. Since $q$ is $(\epsilon, \delta)$-differentially private and the algorithm of Step (2) is only a function of $q$, the composition theorem implies that $h$ is also $(\epsilon, \delta)$-differentially private. Recall that with probability at least $1 - \beta/2$ we have $\|p - \widehat{p}_n\|_{\mathcal{A}_{O(t)}} \leq \alpha/2$. By the properties of $q$ in Step (1), a union bound and an application of the triangle inequality imply that with probability at least $1 - \beta$ we have $\|p - q\|_{\mathcal{A}_{O(t)}} \leq \alpha$. Hence, the output $h$ of Step (2) is an $(\alpha, \beta)$-accurate agnostic hypothesis.

We have thus sketched a proof of the following lemma:

**Lemma 1.** *Suppose there is an $(\epsilon, \delta)$-differentially private synthetic data algorithm under the $\mathcal{A}_{O(t)}$–distance metric that is $(\alpha/2, \beta/2)$-accurate on databases of size $n$, where $n = \Omega((t + \log(1/\beta))/\alpha^2)$. Then, there exists an $(\alpha, \beta)$-accurate agnostic learning algorithm for $\mathcal{H}_t$ with sample complexity $n$.*

Recent work of Bun *et al.* [9] gives an efficient differentially private synthetic data algorithm under the Kolmogorov distance metric:

**Proposition 1.** *[9] There is an $(\epsilon, \delta)$-differentially private $(\alpha, \beta)$-accurate synthetic data algorithm with respect to $d_K$–distance on databases of size $n$ over $[N]$, assuming $n = \Omega((1/(\epsilon\alpha)) \cdot 2^{O(\log^* N)} \cdot \ln(1/\alpha\beta\epsilon\delta))$. The algorithm runs in time $O(n \cdot \log N)$.*

Note that the Kolmogorov distance is equivalent to the $\mathcal{A}_2$-distance up to a factor of 2. Hence, by applying the above proposition for $\alpha' = \alpha/t$ one obtains an $(\alpha, \beta)$-accurate synthetic data algorithm with respect to the $\mathcal{A}_t$-distance. Combining the above, we obtain the following:

**Theorem 5.** *There is an $(\epsilon, \delta)$-differentially private $(\alpha, \beta)$-agnostic learning algorithm for $\mathcal{H}_t$ that uses $n = O((t/\alpha^2) \cdot \ln(1/\beta) + (t/(\epsilon\alpha)) \cdot 2^{O(\log^* N)} \cdot \ln(1/\alpha\beta\epsilon\delta))$ samples and runs in time $\widetilde{O}(n) + O(n \cdot \log N)$.*

As an immediate corollary of Theorem 5, we obtain nearly-sample optimal and computationally efficient differentially private estimators for all the structured discrete distribution families studied

in [3, 4]. These include well-known classes of shape restricted densities including (mixtures of) unimodal and multimodal densities (with unknown mode locations), monotone hazard rate (MHR) and log-concave distributions, and others. Due to space constraints, we do not enumerate the full descriptions of these classes or statements of these results here but instead refer the interested reader to [3, 4].

## 4    Maximum Error Rule for Private Kolmogorov Distance Approximation

In this section, we describe a simple and fast algorithm for privately approximating an input histogram with respect to the Kolmogorov distance. Our private algorithm relies on a simple (non-private) iterative greedy algorithm to approximate a given histogram (empirical distribution) in Kolmogorov distance, which we term MAXIMUMERRORRULE. This algorithm performs a set of basic operations on the data and can be effectively implemented in the private setting.

To describe the non-private version of MAXIMUMERRORRULE, we point out a connection of the Kolmogorov distance approximation problem to the problem of approximating a monotone univariate function with by a piecewise linear function. Let $\widehat{p}_n$ be the empirical probability distribution over $[N]$, and let $\widehat{P}_n$ denote the corresponding empirical CDF. Note that $\widehat{P}_n : [N] \to [0, 1]$ is monotone non-decreasing and piecewise constant with at most $n$ pieces. We would like to approximate $\widehat{p}_n$ by a piecewise uniform distribution with a corresponding a piecewise linear CDF. It is easy to see that this is exactly the problem of approximating a monotone function by a piecewise linear function in $\ell_\infty$-norm.

The MAXIMUMERRORRULE works as follows: Starting with the trivial linear approximation that interpolates between $(0, 0)$ and $(N, 1)$, the algorithm iteratively refines its approximation to the target empirical CDF using a greedy criterion. In each iteration, it finds the point $(x, y)$ of the true curve (empirical CDF $\widehat{P}_n$) at which the current piecewise linear approximation disagrees most strongly with the target CDF (in $\ell_\infty$-norm). It then refines the previous approximation by adding the point $(x, y)$ and interpolating linearly between the new point and the previous two adjacent points of the approximation. See Figure 1 for a graphical illustration of our algorithm. The MAXIMUMERRORRULE is a popular method for monotone curve approximation whose convergence rate has been analyzed under certain assumptions on the structure of the input curve. For example, if the monotone input curve satisfies a Lipschitz condition, it is known that the $\ell_\infty$-error after $T$ iterations scales as $O(1/T^2)$ (see, e.g., [27] and references therein).

There are a number of challenges towards making this algorithm differentially private. The first is that we cannot exactly select the maximum error point. Instead, we can only choose an approximate maximizer using a differentially private sub-routine. The standard algorithm for choosing such a point would be the exponential mechanism of McSherry and Talwar [28]. Unfortunately, this algorithm falls short of our goals in two respects. First, it introduces a linear dependence on the domain size in the running time making the algorithm prohibitively inefficient over large domains. Second, it introduces a logarithmic dependence on the domain size in the error of our approximation.

In place of the exponential mechanism, we design a sub-routine using the "choosing mechanism" of Beimel, Nissim, and Stemmer [25]. Our sub-routine runs in logarithmic time in the domain size and achieves a doubly-logarithmic dependence in the error. See Figure 2 for a pseudocode of our algorithm. In the description of the algorithm, we think of $A_t$ as a CDF defined by a sequence of points $(0, 0), (x_1, y_1), ..., (x_k, y_k), (N, 1)$ specifying the value of the CDF at various discrete points of the domain. We denote by $\text{weight}(I, A_t) \in [0, 1]$ the weight of the interval $I$ according to the CDF $A_t$, where the value at missing points in the domain is achieved by linear interpolation. In other words, $A_t$ represents a piecewise-linear CDF (corresponding to a piecewise constant histogram). Similarly, we let $\text{weight}(I, S) \in [0, 1]$ denote the weight of interval $I$ according to the sample $S$, that is, $|S \cap I|/|S|$.

We show that our algorithm satisfies $(\epsilon, \delta)$-differential privacy (see Appendix B):

**Lemma 2.** *For every $\epsilon \in (0, 2), \delta > 0$, MaximumErrorRule satisfies $(\epsilon, \delta)$-differential privacy.*

Next, we provide two performance guarantees for our algorithm. The first shows that the running time per iteration is at most $O(n \log N)$. The second shows that if at any step $t$ there is a "bad" interval in $\mathcal{I}$ that has large error, then our algorithm finds such a bad interval where the quantitative

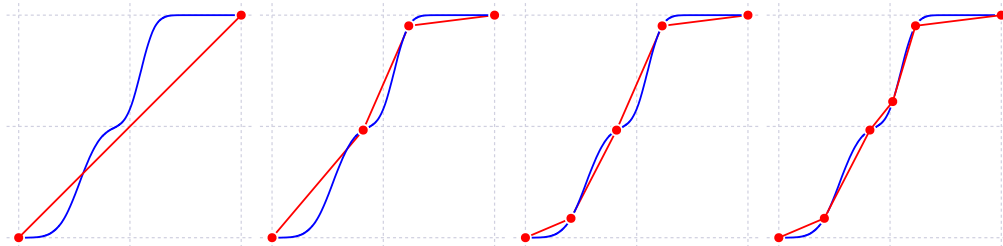

Figure 1: CDF approximation after $T = 0, 1, 2, 3$ iterations.

---

MAXIMUMERRORRULE($S \in [N]^n$, privacy parameters $\epsilon, \delta$)
For $t = 1$ to $T$ :
      1. $I = $ FINDBADINTERVAL($A_{t-1}, S$)
      2. $A_t = $ UPDATE($A_{t-1}, S, I$)

FINDBADINTERVAL
      1. Let $\mathcal{I}$ be the collection of all dyadic intervals of the domain.
      2. For each $J \in \mathcal{I}$, let $q(J; S) = |\text{weight}(J, A_{t-1}) - \text{weight}(J, S)|$.
      3. Output an $I \in \mathcal{I}$ sampled from the choosing mechanism with score function $q$ over the collection $\mathcal{I}$ with privacy parameters $(\epsilon/2T, \delta/T)$.

UPDATE
      1. Let $I = (l, r)$ be the input interval. Compute $w_l = \text{weight}([1, l], S) + \text{Laplace}(0, 1/(2n\epsilon))$ and $w_r = \text{weight}([l+1, r], S) + \text{Laplace}(0, 1/(2n\epsilon))$.
      2. Output the CDF obtained from $A_{t-1}$ by adding the points $(l, w_l)$ and $(r, w_l + w_r)$ to the graph of $A_{t-1}$.

Figure 2: Maximum Error Rule (MERR).

loss depends only doubly-logarithmically on the domain size (see Appendix B for the proof of the following theorem).

**Proposition 2.** *MERR runs in time $O(Tn \log N)$. Furthermore, for every step $t$, with probability $1 - \beta$, we have that the interval $I$ selected at step $t$ satisfies*

$$|\text{weight}(I, A_{t-1}) - \text{weight}(I, S)| \geq \text{OPT} - O\left(\frac{1}{\epsilon n} \cdot \log\left(n \log N \cdot \log(1/\beta\epsilon\delta)\right)\right).$$

*Recall that* $\text{OPT} = \max_{J \in \mathcal{I}} |\text{weight}(J, A_{t-1}) - \text{weight}(J, S)|$.

## 5 Experiments

In addition to our theoretical results from the previous sections, we also investigate the empirical performance of our private distribution learning algorithm based on the maximum error rule. The focus of our experiments is the learning error achieved by the private algorithm for various distributions. For this, we employ two types of data sets: multiple synthetic data sets derived from mixtures of well-known distributions (see Appendix C), and a data set from Higgs experiments [29]. The synthetic data sets allow us to vary a single parameter (in particular, the domain size) while keeping the remaining problem parameters constant. We have chosen a distribution from the Higgs data set because it gives rise to a large domain size. Our results show that the maximum error rule finds a good approximation of the underlying distribution, matching the learning error of a non-private baseline when the number of samples is sufficiently large. Moreover, our algorithm is very efficient and runs in less than 5 seconds for $n = 10^7$ samples on a domain of size $N = 10^{18}$.

We implemented our algorithm in the Julia programming language (v0.3) and ran the experiments on an Intel Core i5-4690K CPU (3.5 - 3.9 GHz, 6 MB cache). In all experiments involving our private learning algorithm, we set the privacy parameters to $\epsilon = 1$ and $\delta = \frac{1}{n}$. Since the noise magnitude depends on $\frac{1}{\epsilon n}$, varying $\epsilon$ has the same effect as varying the the sample size $n$. Similarly, changes in $\delta$ are related to changes in $n$, and therefore we only consider this setting of privacy parameters.

**Higgs data.**    In addition to the synthetic data mentioned above, we use the lepton $p_T$ (transverse momentum) feature of the Higgs data set (see Figure 2e of [29]). The data set contains roughly 11 million samples, which we use as unknown distribution. Since the values are specified with 18 digits of accuracy, we interpret them as discrete values in $[N]$ for $N = 10^{18}$. We then generate a sample from this data set by taking the first $n$ samples and pass this subset as input to our private distribution learning algorithm. This time, we measure the error as Kolmogorov distance between the hypothesis returned by our algorithm and the cdf given by the full set of 11 million samples.

In this experiment (Figure 3), we again see that the maximum-error rule achieves a good learning error. Moreover, we investigate the following two aspects of the algorithm: (i) The number of steps taken by the maximum error rule influences the learning error. In particular, a smaller number of steps leads to a better approximation for small values of $n$, while more samples allow us to achieve a better error with a larger number of steps. (ii) Our algorithm is very efficient. Even for the largest sample size $n = 10^7$ and the largest number of MERR steps, our algorithm runs in less than 5 seconds. Note that on the same machine, simply sorting $n = 10^7$ floating point numbers takes about 0.6 seconds. Since our algorithm involves a sorting step, this shows that the overhead added by the maximum error rule is only about $7\times$ compared to sorting. In particular, this implies that *no* algorithm that relies on sorted samples can outperform our algorithm by a large margin.

**Limitations and future work.**    As we previously saw, the performance of the algorithm varies with the number of iterations. Currently this is a parameter that must be optimized over separately, for example, by choosing the best run privately from the exponential mechanism. This is standard practice in the privacy literature, but it would be more appealing to find an adaptive method of choosing this parameter on the fly as the algorithm obtains more information about the data.

There remains a gap in sample complexity between the private and the non-private algorithm. One reason for this are the relatively large constants in the privacy analysis of the choosing mechanism [9]. With a tighter privacy analysis, one could hope to reduce the sample size requirements of our algorithm by up to an order of magnitude.

It is likely that our algorithm could also benefit from certain post-processing steps such as smoothing the output histogram. We did not evaluate such techniques here for simplicity and clarity of the experiments, but this is a promising direction.

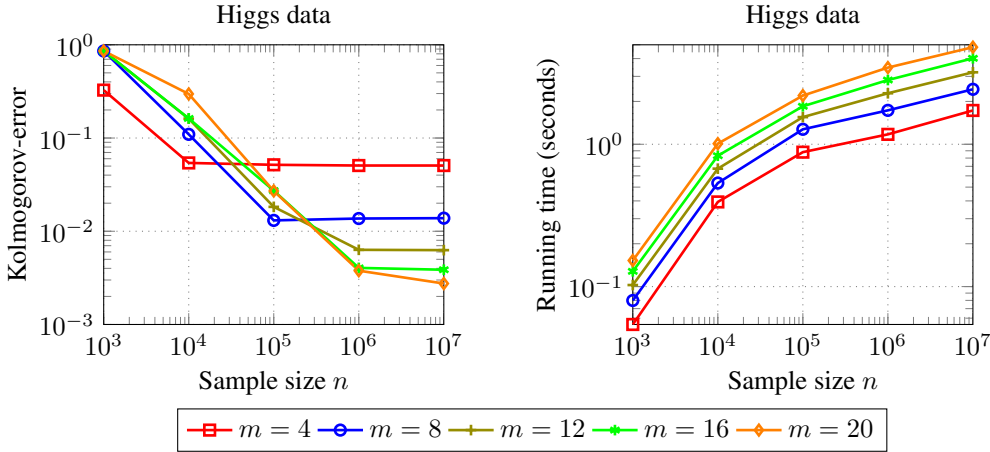

Figure 3: Evaluation of our private learning algorithm on the Higgs data set. The left plot shows the Kolmogorov error achieved for various sample sizes $n$ and number of steps taken by the maximum error rule ($m$). The right plot displays the corresponding running times of our algorithm.

## Acknowledgments

Ilias Diakonikolas was supported by EPSRC grant EP/L021749/1 and a Marie Curie Career Integration grant. Ludwig Schmidt was supported by MADALGO and a grant from the MIT-Shell Initiative.

## Footnotes

[1] A statistical query asks for the average of a predicate over the dataset.

[2]We remark that a potential difference is in the running time of the algorithm, which depends on the support and structure of the distribution $q$.

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
