[Supplementary Material]

# A   Omitted Proofs from Section 3

## A.1   Proof of Theorem 3

The algorithm proceeds as follows: Given a dataset $S$ of $n$ samples from the target distribution $p$ over $[N]$, it outputs the hypothesis $h = \text{hist}(S) + \eta = \widehat{p}_n(S) + \eta$, where $\eta \in \mathbb{R}^N$ is sampled from the $N$-dimensional Laplace distribution with standard deviation $1/(\epsilon n)$.

It is easy to see that this algorithm is $(\epsilon, 0)$-differentially private, so it remains to prove that it is $(\alpha, \beta)$-accurate. By properties of the Laplace distribution, it follows that with probability $1 - \beta/2$ we have $\|\eta\|_\infty \leq O(\log(1/\beta)/(\epsilon n))$. Hence,
$$\|\eta\|_1 \leq O(N \log(1/\beta)/\epsilon n) \leq \alpha \ .$$
Moreover, by combining Theorems 1 and 2 we obtain that with probability at least $1 - \beta/2$ it holds $d_{TV}(\widehat{p}_n, p) \leq \alpha/2$. By a union bound we conclude that with probability at least $1 - \beta$
$$d_{TV}(h, p) \leq d_{TV}(\widehat{p}_n, p) + \frac{1}{2}\|\eta\|_1 \leq \alpha \ ,$$
which completes the proof.

## A.2   Proof of Theorem 4

The main idea of the proof is that the differentially private learning problem for $\mathcal{H}_t(\mathcal{I})$ can be effectively reduced to the same problem over distributions of support $[t]$. The theorem then follows by an application of Theorem 3. To formally establish this intuitive claim, we need a couple of definitions. Let $p$ be the unknown target distribution over $[N]$ and $\mathcal{I} = \{I_i\}_{i=1}^t$ be the known partition of $[N]$ into $t$ intervals. We remark that $p$ is potentially arbitrary, and in particular it is *not* necessarily the case that $p \in \mathcal{H}_t(\mathcal{I})$.

The *flattened distribution* $(p_f)^{\mathcal{I}}$ corresponding to $p$ and $\mathcal{I}$ is the distribution over $[N]$ defined as follows: for $j \in [t]$ and $i \in I_j$, $(p_f)^{\mathcal{I}}(i) = p(I_j)/|I_j|$. That is, $(p_f)^{\mathcal{I}}$ is obtained from $p$ by averaging the weight that $p$ assigns to each interval over the entire interval. The *reduced distribution* $(p_r)^{\mathcal{I}}$ corresponding to $p$ and $\mathcal{I}$ is the distribution over $[t]$ that assigns the $i$-th point the weight $p$ assigns to the interval $I_i$; i.e., for $j \in [t]$, we have $(p_r)^{\mathcal{I}}(j) = p(I_j)$.

Since the partition $\mathcal{I}$ is explicitly known to the algorithm, given a sample from $p$ it is straightforward to simulate a sample from $(p_r)^{\mathcal{I}}$. In view of this observation, our algorithm proceeds as follows: We use samples from $p$ to simulate samples from $(p_r)^{\mathcal{I}}$. Since $(p_r)^{\mathcal{I}}$ is supported on $[t]$, we can use the algorithm of Theorem 3 to obtain an $(\epsilon, 0)$-differentially private $(\alpha, \beta)$-learning algorithm for $(p_r)^{\mathcal{I}}$ using the desired number of samples. Let $(h_r)^{\mathcal{I}}$ be the output of the algorithm of Theorem 3. Our hypothesis $h$ is obtained by averaging the weight that $(h_r)^{\mathcal{I}}$ assigns to its $j$-th point over the corresponding interval $I_j$, i.e., for $j \in [t]$ and $i \in I_j$, $h(i) = (h_r)^{\mathcal{I}}(j)/|I_j|$.

The sample size and running time of our algorithm follows from Theorem 3. Since the output of the first step, i.e., $(h_r)^{\mathcal{I}}$, is $(\epsilon, 0)$-differentially private and the second step simply post-processes $(h_r)^{\mathcal{I}}$ to obtain $h$, by the composition of differential privacy our overall algorithm is $(\epsilon, 0)$-differentially private. It remains to show that our algorithm is an $(\alpha, \beta)$-agnostic learning algorithm. We prove this in the following paragraph.

Observe that for any pair of distributions $p, q$ and any partition $\mathcal{I}$ of $[N]$ into intervals, we have that $d_{TV}((p_r)^{\mathcal{I}}, (q_r)^{\mathcal{I}}) = d_{TV}((p_f)^{\mathcal{I}}, (q_f)^{\mathcal{I}})$. Since $h \in \mathcal{H}_t(\mathcal{I})$, this yields $d_{TV}((p_r)^{\mathcal{I}}, (h_r)^{\mathcal{I}}) = d_{TV}((p_f)^{\mathcal{I}}, h)$. We thus have that $\left|d_{TV}(p, h) - d_{TV}((p_r)^{\mathcal{I}}, (h_r)^{\mathcal{I}})\right|$ is equal to
$$\left|d_{TV}(p, h) - d_{TV}((p_f)^{\mathcal{I}}, h)\right| = d_{TV}(p, h) - d_{TV}((p_f)^{\mathcal{I}}, h) \leq d_{TV}(p, (p_f)^{\mathcal{I}}),$$
where the equality above is equivalent to $d_{TV}(p, h) \geq d_{TV}((p_f)^{\mathcal{I}}, h)$ (which is easily verified by considering each interval $I_i \in \mathcal{I}$ separately and applying triangle inequality) and the inequality is the triangle inequality. Since $\text{opt}_t(p) = \inf_{g \in \mathcal{H}_t(\mathcal{I})} d_{TV}(g, p)$, it follows that there exists $g^* \in \mathcal{H}_t(\mathcal{I})$ such that $d_{TV}(g^*, p) = \text{opt}_t(p)$. This can be shown to imply that $d_{TV}(p, (p_f)^{\mathcal{I}}) \leq 2\text{opt}_t(p)$ . The RHS above is thus bounded by $2\text{opt}_t(p)$. By Theorem 3, with probability at least $1 - \beta$ it holds $d_{TV}((p_r)^{\mathcal{I}}, (h_r)^{\mathcal{I}}) \leq \alpha$, and therefore
$$d_{TV}(p, h) \leq d_{TV}((p_r)^{\mathcal{I}}, (h_r)^{\mathcal{I}}) + d_{TV}(p, (p_f)^{\mathcal{I}}) \leq \alpha + 2\text{opt}_t(p).$$
This completes the proof.

CHOOSINGMECHANISM($S \in [N]^n$, privacy parameters $\epsilon, \delta$, failure probability $\beta$)
1. Let OPT $= \max_{I \in \mathcal{I}} q(I; S)$. Compute $\widetilde{\text{OPT}} = \text{OPT} + \text{Laplace}(4/\epsilon n)$. If $\widetilde{\text{OPT}} < \frac{8}{\epsilon n} \ln(8\lceil \log N \rceil / \beta \delta)$, halt and return $\bot$.
2. Let $G(S) = \{S \in \mathcal{I} : q(J; S) \geq 1\}$. Sample $I \in \mathcal{I}$ from the exponential mechanism with privacy parameter $\epsilon$, that is, according to the density $Z e^{\epsilon q(I;S)/2}$, where $Z$ is the appropriate normalization constant.

Figure 4: Choosing mechanism for the Maximum Error Rule (MERR)

# B    Omitted Proofs from Section 4

**Lemma 2.** *For every $\epsilon \in (0, 2), \delta > 0$, MaximumErrorRule satisfies $(\epsilon, \delta)$-differential privacy.*

*Proof.* To prove the claim we first note that the choosing mechanism as described in Bun et al. [9] requires a parameter $B$ such that the score function $q(J; S)$ satisfies a technical condition called "$B$-bounded growth". This condition enforces that a change in one element in $S$ can change the score function for at most $B$ choices of $S$. Moreover, the score function must be $(1/n)$-sensitive, meaning that a change to one element in $S$ can change any single score by at most $(1/n)$. We implicitly set the growth parameter to $B = 2\lceil \log(N) \rceil$ and this is justified for the following reason. A single data point is contained in at most $B/2$ dyadic intervals. Hence, replacing on data point by another can affect the score of at most $B$ intervals. We can now make use of the following lemma:

**Lemma 3** ([25, 9]). *Fix $\delta > 0$ and $0 < \epsilon \leq 2$. If $q$ is a $B$-bounded growth score function, then the choosing mechanism is $(\epsilon, \delta)$-differentially private.*

By the previous lemma, each invocation of the choosing mechanism satisfies $(\epsilon/2T, \delta/T)$-differential privacy. We claim that each update step satisfies $(\epsilon/2, 0)$-differential privacy. This follows because we evaluate the weight on two disjoint intervals. A single data point can only participate in one of the two intervals and therefore, the $\ell_1$-sensitivity of the pair $(\text{weight}([1, l], S), \text{weight}([l + 1, r], S)$ is bounded by $1/n$. Hence, adding Laplacian noise of magnitude $1/(2n\epsilon)$ to each number ensures $(\epsilon/2, 0)$-differential privacy. By the basic composition rule for differential privacy, each step of our algorithm satisfies $(\epsilon/T, \delta/T)$-differential privacy so that the entire algorithm satisfies $(\epsilon, \delta)$-differential privacy. $\square$

We note that we could have alternatively applied the advanced composition rule [30] to obtain a better dependence on $T$ in setting the privacy parameter in each step (namely, $O(\sqrt{T})$ rather than $T$). However, the algorithm typically converges in a very small number of steps so that this setting leads to worse empirical performance due to the overhead of the advanced composition rule in terms of constants and an extra factor of $\log(1/\delta)$.

**Proposition 2.** *MERR runs in time $O(Tn \log N)$. Furthermore, for every step $t$, with probability $1 - \beta$, we have that the interval $I$ selected at step $t$ satisfies*

$$|\text{weight}(I, A_{t-1}) - \text{weight}(I, S)| \geq \text{OPT} - O\left(\frac{1}{\epsilon n} \cdot \log\left(n \log N \cdot \log(1/\beta \epsilon \delta)\right)\right).$$

*Recall that* $\text{OPT} = \max_{J \in \mathcal{I}} |\text{weight}(J, A_{t-1}) - \text{weight}(J, S)|$.

*Proof.* The claim about run time is straightforward to prove. While there are $O(N \log N)$ dyadic intervals, we only need to compute the weight for those that contain at least one data point. This can be done in time $O(n \log N)$ per iteration.

The second claim follows from Lemma 3.8 in [9]. $\square$

# C    Additional Experiments

In our synthetic data experiments, we start with a truncated continuous mixture distribution of Gaussian, Beta, or Gamma components (see the first row in Figure 5 for the corresponding densities). We

then convert this truncated distribution to a discrete distribution over a given domain size $N$. In each run of the experiment, we draw $n$ samples and pass these samples as input to the non-private histogram learning algorithm of [4] and our private learning algorithm. As parameters, we choose $k = 20$ histogram bins and $m = 20$ steps of the maximum error rule. The final error metric is the $\ell_1$-distance between the hypothesis produced by the algorithms and the true underlying distribution.

The experiments show two main points. (i) There is a price for privacy, but more samples can compensate it: in the regime of smaller values of $n$, the $\ell_1$-learning error achieved by the private algorithm is worse than the non-private counterpart (second row of Figure 5). However, for larger values of $n$, the private algorithm achieves the same learning error as the non-private histogram learning algorithm. (ii) As predicted by our theoretical analysis, row three of Figure 5 shows that the learning error of our algorithm is essentially independent of the domain size. The maximum error rule achieves the same learning error for domain sizes ranging from $N = 10^6$ to $N = 10^{10}$.

Figure 5: Evaluation of our private distribution learning algorithm on synthetic data. Row 1: The three test distributions. Row 2: Private vs. non-private learning error. Row 3: Private learning error for various domain sizes. Every data point is averaged over ten independent trials.