[Reviews · NeurIPS 2015]

Submitted by Assigned_Reviewer_1

The paper has a nice idea, and has some nice theory and experiments. There are perhaps two limitations. First, the problem is a little limited -- the kind of distributions that are being described are essentially low-dimensional. Second, the paper could do with some more discussion of related work, especially those in the NIPS/ICML community. Important citations that are missing are [1] and [2]: lower bounds on range approximation with pure differential privacy were shown by [1], and density estimation via differentially private histograms (via the L2 distance) has been done by [2].

[1] Convergence Rates of Differentially Private Statistical Estimation, Chaudhuri and Hsu, ICML 2012 [2] Differentially Private M-Estimators, Lei, NIPS 2011
Summary: This paper provides an algorithm for learning distributions which are polynomially approximable with differential privacy based on samples. The main contribution of this paper is to construct differentially private approximations to histograms in Kolmogorov distance, and then use these private histograms for density estimation. The idea is nice, and could have further practical applications. Overall this is a good paper, and I would recommend acceptance.

Submitted by Assigned_Reviewer_2

This paper studies the problem of learning discrete probability distributions over large finite sets under privacy constraints. The contributions are split into three parts: a theoretical study with formal accuracy, privacy, and computational guarantees; a fast empirical algorithm with privacy guarantees; and, a set of experiments on synthetic data and a real dataset.

Broadly speaking, the main idea of the paper is to leverage previous work on algorithms for learning adaptive one-dimensional histograms over large sets and combine them with noise-generating procedures to obtain private algorithms. This is an important theoretical problem, and the authors manage to show in Section 3 that privacy can be achieved at a very reasonable price in terms of statistical accuracy. The techniques involved in Theorems 1 and 2 are not especially novel: they rely on the Laplace mechanism and a simple reduction. Theorem 3 is more interesting, but two important details were missing from the original submission. These were provided in the rebuttal and, though they are relatively simple, my suggestion is the authors incorporate them in the supplementary material in further versions of the paper.

The practical impact of the contributions presented in the paper might be limited. The authors mention several families of univariate discrete distributions which can be agnostically learned using their techniques. These are quite broad, and seem obviously related to the cases where a data analyst would benefit from a histogram representation of the target distribution. However, the one-dimensional assumptions seems a rather strong assumption for many practical applications. If one had access to samples from a high-dimensional discrete distribution, it does not seem possible that the algorithms presented in the paper can be useful in extracting relevant information about the geometry of such distribution, since different projections to a one-dimensional setting would probably yield quite different histograms. It would be nice if the authors commented on these limitations and whether there are obvious extensions of their techniques for such cases, or on the other hand one would need completely different methods.

*** Detalied Comments and Typos ***

- In general, I think the material in Section 3 would benefit from some restructuring where algorithms and analyses are decoupled; after all, statements of the form "there exists an algorithm for doing that and this" are less satisfying that statements about a clearly specified algorithm.

- According to Definition 1, an (alpha,beta)-agnostic learning algorithm produces a "hypothesis distribution h". However, the output of the algorithm analyzed in Theorem 1 might not be a distribution due to the random perturbation. Therefore, this algorithm is not strictly an (alpha,beta)-agnostic learning algorithm in the sense of Definition 1. This mismatch should be discussed and/or fixed.

- Citations for the results mentioned in Lines 60-62 are missing.

- When Theorems 4 and 5 are cited in Line 183, please say they can be found in the supplementary material. Though most theory conferences accept long papers describing their essential contributions in the first pages, NIPS still has a clear division between main paper and supplementary material, and the former is supposed to be self-contained.

- Lemma 1 should say that the algorithm in the thesis is also private. Furthermore, the relevance of requiring in the hypothesis of the lemma that the first algorithm has a particular sample complexity is not entirely clear.

- Non-increasing -> Non-decreasing (Line 240)

- \hat{P}_n -> \hat{p}_n (Line 240)

- distance Combining (Line 215)

- curve For example (Line 251)
Summary: Important results about a theoretically relevant problem; practical impact is unclear. There is room for improvement in the presentation.

Submitted by Assigned_Reviewer_3

This study investigates differentially private estimation of distributions over discrete domains. For the class of structured distributions, by making use of the algorithmic framework introduced by [4], the proposed scheme is a sample and computationally efficient private estimation algorithm in terms of the total variation. The authors showed that the proposed scheme was nearly as efficient as a nearly optimal non-private analogue, both in terms of time and sample size.

Most of the background is deferred to the supplementary, and this might be inconvenient to readers. Most of the readers of this paper might not need the definition of differential privacy. However, Theorem 3, Theorem 4, and "connection to synthetic data" parts seems to be necessary to understand the whole idea.

Is "near sample optimality" of Theorem 3 at Line 220 is claimed by the upper bound of the sample complexity shown by Theorems 4 and Theorem 5 as Theta((t+log(1/beta))/alpha^2)? If so, I am not convinced that the proposed scheme achieves nearly sample optimality by Theorem 3. I doubt that the upper bound of the sample complexity is satisfied for (epsilon, delta)-differentially private (alpha, beta)-accurate agnostic learning algorithm for {\cal H}_t. The sentence at Lines 050-051, "nearly as efficient-both in terms of sample size and running time-as a nearly optimal non-private baseline", might be your precise claim for optimality.

The organization of the entire paper is not self-consistent. I think it is almost impossible to understand the idea without reading the supplementary.

This study is composed of known results (e.g., distribution learning in [4], choosing mechanism in [25], data synthesizer in [9], etc.). In this sense, the originality of contribution is not very high, but the study entirely derives interesting conclusions and is worth publishing at NIPS.
Summary: This study is composed of known results (e.g., distribution learning in [4], choosing mechanism in [25], data synthesizer in [9], etc.). In this sense, the originality of contribution is not very high, but the study entirely derives interesting conclusions and is worth publishing at NIPS.

Author Feedback
Author rebuttal: We thank the reviewers for their valuable comments.

Reviewer_1:

- Since our paper relies on multiple recent results, describing each technique in detail would have filled a large fraction of the paper. Nevertheless, we take this point seriously and will make the paper more self-contained.

- We use the term "synthetic data algorithm" for an algorithm that takes an arbitrary database as input and produces an output database with certain desirable properties. In particular, Lemma 1 and Proposition 1 require that this algorithm is differentially private and approximately preserves the distribution induced by the input database.

Reviewer_2:

- We emphasize that the proofs given by us are complete and correct. In particular, we now address the two points raised by the reviewer.

- In Line 184 (and essentially all of the paper), A_t refers to the family of sets that can be written as the union of t disjoint intervals. We did not explicitly state the VC dimension of A_t since its derivation is similar to that of a single interval, which is a common textbook example. To avoid any misunderstandings, we now give a more detailed explanation:

Let N = 2t + 1 and X = [N], using the notation from line 511. Moreover, let Y be the set of odd integers up to 2t + 1. Note that Y consists of t + 1 disjoint intervals. Since every set in the family A_t consists of only t disjoint intervals, A_t does not shatter X and hence the VC-dimension of A_t is at most 2t. It is easy to see (and well-known) that every set of size at most 2t can be shattered by A_t, regardless of N. So the VC-dimension of A_t is 2t.

- Line 215: the claim follows directly from the definition of the A_k-distance. To illustrate this, we now formally show that accuracy alpha / t in the A_2-distance implies accuracy alpha in the A_t distance.

Let f be a function with || f ||_{A_2} <= alpha / t, and let J be an arbitrary collection of t disjoint intervals I_1, ..., I_t. We have to show that | f(I_1) | + ... + | f(I_t) | <= alpha. For each i, the bound on the A_2-norm of f implies | f(I_i) | <= alpha / t. Substituting this into the expression above proves the claim.

- Regarding practicality, we point out that univariate distributions are already interesting in applications and have received much attention in the privacy and density estimation literatures (see references). E.g., much of the work on range queries in differential privacy (see [24]) has a very practical motivation. However, almost all of these algorithms have a polynomial dependence on the domain size N in the running time or a bad dependence in the error, which can be prohibitive for large data sets. To the best of our knowledge, our algorithm is the first implemented algorithm with a sub-logarithmic dependence on the domain size in the error and a logarithmic dependence in running time, which is an important step towards practicality.

Moreover, it is possible to map a high-dimensional distribution to an exponentially-sized univariate representation. This is a common step for many algorithms in differential privacy (such as the well-known private multiplicative weights algorithm). But most methods then run in time exponential in the number of attributes, while our algorithm continues to run efficiently due to its sub-logarithmic dependence on the domain size. Hence our algorithm is still a valuable heuristic for high-dimensional problems.

Finally, once algorithms with guarantees similar to [4] and [9] are developed for multidimensional histograms / synthetic data, combining these methods with our work will immediately yield private algorithms for learning multi-dimensional structured distributions.

- As stated, the hypothesis h in Theorem 1 is not guaranteed to be a probability distribution. However, it is close to a distribution because of the bound on the l1-norm of eta. By adjusting the probabilities so that they are nonnegative and sum to 1, we get a true distribution with the same accuracy guarantees.

- Lemma 1 should indeed state that the resulting algorithm is (eps, delta)-private.

The lemma requires a lower bound on the sample complexity to guarantee that the empirical distribution is close to the unknown distribution in A_t-distance. Combined with the accuracy guarantee of the private synthetic data algorithm, this implies that the output of the private synthetic data algorithm (and hence the input to the non-private learning algorithm) is also close to the unknown distribution.

Reviewer_3:

- Regarding our claim to optimality: Our private upper bound nearly matches a non-private lower bound. Since the private learning task is only harder than the non-private learning task, the existing lower bound also applies to private learning. Hence, our private upper bound is within logarithmic factors of the private lower bound.

Reviewer_5:

- Regarding low-dimensionality, please see our response to Reviewer 2.